# MRI overestimates articular cartilage thickness and volume compared to synchrotron radiation phase-contrast imaging

Suranjan Bairagi[1], Mohammad-Amin Abdollahifar[1], Oghenevwogaga J. Atake[1], William Dust[2], Sheldon Wiebe[3], George Belev[4], L. Dean Chapman[1], M. Adam Webb[4], Ning Zhu[4], David M. L. Cooper[1], B. Frank Eames[1]*

1 Department of Anatomy, Physiology, and Pharmacology, University of Saskatchewan, Saskatoon, Saskatchewan, Canada, 2 Department of Surgery, University of Saskatchewan, Saskatoon, Saskatchewan, Canada, 3 Department of Medical Imaging, University of Saskatchewan, Saskatoon, Saskatchewan, Canada, 4 Canadian Light Source Inc., Saskatoon, Saskatchewan, Canada

* b.frank@usask.ca

**Data Availability Statement:** In order that others may replicate or extend our study findings, raw image files from all samples can be found at https://globus.usask.ca/file-manager?origin_id=

## Abstract

Accurate evaluation of morphological changes in articular cartilage are necessary for early detection of osteoarthritis (OA). 3T magnetic resonance imaging (MRI) has highly sensitive contrast resolution and is widely used clinically to detect OA. However, synchrotron radiation phase-contrast imaging computed tomography (SR-PCI) can also provide contrast to tissue interfaces that do not have sufficient absorption differences, with the added benefit of very high spatial resolution. Here, MRI was compared with SR-PCI for quantitative evaluation of human articular cartilage. Medial tibial condyles were harvested from non-OA donors and from OA patients receiving knee replacement surgery. Both imaging methods revealed that average cartilage thickness and cartilage volume were significantly reduced in the OA group, compared to the non-OA group. When comparing modalities, the superior resolution of SR-PCI enabled more precise mapping of the cartilage surface relative to MRI. As a result, MRI showed significantly higher average cartilage thickness and cartilage volume, compared to SR-PCI. These data highlight the potential for high-resolution imaging of articular cartilage using SR-PCI as a solution for early OA diagnosis. Recognizing current limitations of using a synchrotron for clinical imaging, we discuss its nascent utility for preclinical models, particularly longitudinal studies of live animal models of OA.

## Introduction

The most common cartilage disease is osteoarthritis (OA), where articular cartilage of the joint is destroyed either as a normal aging process or as a secondary disease from trauma [1, 2]. Given associated pain, swelling, and joint dysfunction, OA dramatically affects work and life productivity. Although destruction of articular cartilage is a gradual process, OA is hard to detect early, delaying diagnosis until significant patient symptoms exist and irreversible joint damage has occurred [1, 2]. As a result, most OA diagnoses are severe cases and are treated by replacing the joint surfaces with prosthetic components. The ability to accurately assess

ef36db6f-8e10-4b8b-93b5-ec8e895a7511&origin_path=%2F.

**Funding:** This study was supported by grants to BFE: Royal University Hospital (RUH) Foundation, the Saskatchewan Health Research Foundation (SHRF) Establishment Grant, and the Canadian Institutes of Health Research (CIHR) project grant 148683. The funders had no role in study design, data collection and analysis, decision to publish, or preparation of the manuscript. Part of the research described in this paper was performed at the Canadian Light Source, a national research facility of the University of Saskatchewan, which is supported by the Canada Foundation for Innovation (CFI), the Natural Sciences and Engineering Research Council (NSERC), the National Research Council (NRC), CIHR, the Government of Saskatchewan, and the University of Saskatchewan.

**Competing interests:** The authors have declared that no competing interests exist.

structural changes in articular cartilage would enable early detection of OA, opening up new avenues of treatment that might prevent disease progression [1, 3–5].

Common clinical techniques to image articular cartilage make OA challenging to diagnose [1, 6–9]. While standard X-ray radiographs are the most common type of imaging in clinics, cartilage is not visible with this modality, so one is left assessing joint space narrowing as a surrogate for cartilage loss. Clearly, standard X-rays cannot detect early or minor changes to articular cartilage [10–12].

Currently, the preferred clinical modality to image cartilage is magnetic resonance imaging (MRI) [13–16]. MRI uses a large magnetic field and radio waves to coordinate the nuclear magnetization of hydrogen atoms [14]. The contrast in MRI is based on the differences of hydrogen (protons) in the various tissues, such as cartilage, compared to surrounding tissues [15]. Accordingly, several MRI protocols for imaging articular cartilage semi-quantitatively and quantitatively have been developed in clinical research studies, each having strengths and weaknesses [17–20]. Two common clinically-relevant parameters, average cartilage thickness and overall cartilage volume, can be measured from MRI datasets using a fully automatic method for subregional segmentation of articular cartilage [21, 22]. For example, MRI datasets were used to determine the extent to which anatomical force application explained variation in femoral and tibial cartilage thickness and volume in knee OA [23].

Although higher strength magnets can increase MRI resolution by detecting signals in smaller voxel sizes, 3 Tesla (3T) scanners are the preferred strength in most musculoskeletal imaging clinics [24]. 3T MRI has limited resolution to visualize details of cartilage morphology [25–27]. For example, knee OA MRI studies showed an overall mean error of ±0.2–0.3 mm in cartilage thickness, an inaccuracy that could stand in the way of early OA diagnosis [28, 29]. Therefore, to achieve a more accurate diagnosis of morphological changes in articular cartilage, more sensitive imaging technologies should be developed [30, 31].

One unique imaging method is synchrotron radiation phase-contrast imaging computed tomography (SR-PCI). In a synchrotron, high-energy electromagnetic radiation is emitted by electrons moving at speeds close to that of light [32, 33]. SR enables high-resolution imaging because it has high photon flux, stable sources, and an energy density 1000 times higher than conventional X-ray absorption imaging [34, 35]. Some synchrotron imaging techniques take advantage of the high resolution, including absorption-based micro-CT, or even revealing material properties by small angle X-ray scattering [36, 37]. However, the coherent X-rays in SR can reveal phase contrast depending on the refractive index of different tissues, and diffraction-enhanced imaging is even possible [38]. The easiest phase contrast set-up in a synchrotron is where the sample is in-line with the beamline and detector [34, 35, 39]. As a result of these features, SR-PCI enables high-resolution imaging of soft tissues, such as articular cartilage, potentially providing more accurate information about cartilage degeneration in OA [31, 35].

Here, we focused on two main research objectives. First, can SR-PCI generate sufficient data to quantitate clinically-relevant articular cartilage parameters? Second, how do quantitative image analyses from 3T MRI compare to SR-PCI? Previous proof-of-principle papers using single samples compared SR-PCI to 3T MRI [24, 30, 37, 38]. Here, for the first time, measurements of multiple samples of human articular cartilage from medial tibial condyles of non-OA donors and OA patients permit statistical analyses to compare these imaging modalities.

## Materials and methods

### Ethics statement and sample collection and preparation

All tissues were obtained according to University of Saskatchewan-approved ethical protocol Bio 13–110. Healthy donor tissues were harvested through the Body Bequeathal Program of

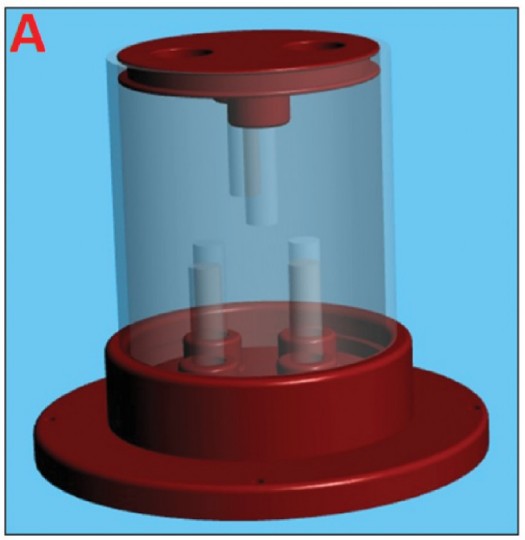

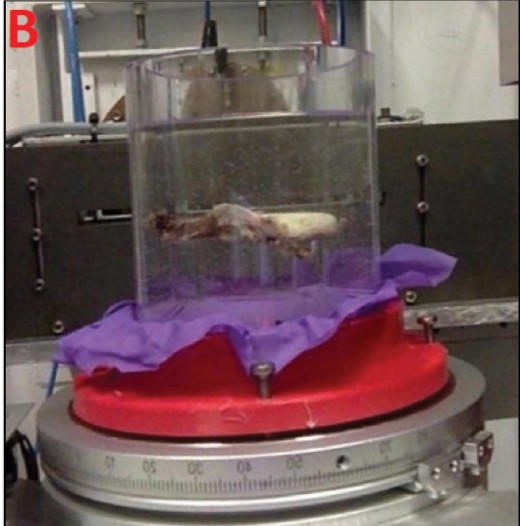

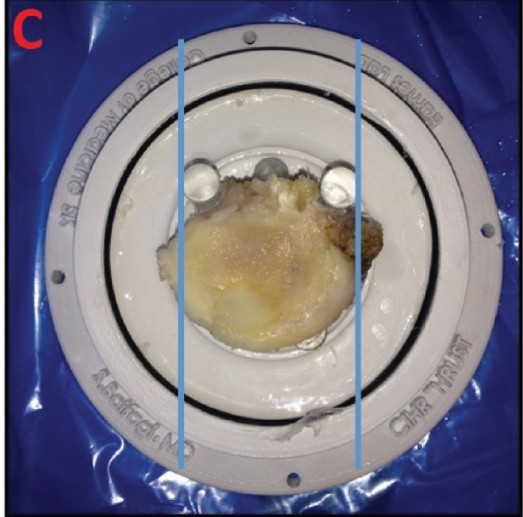

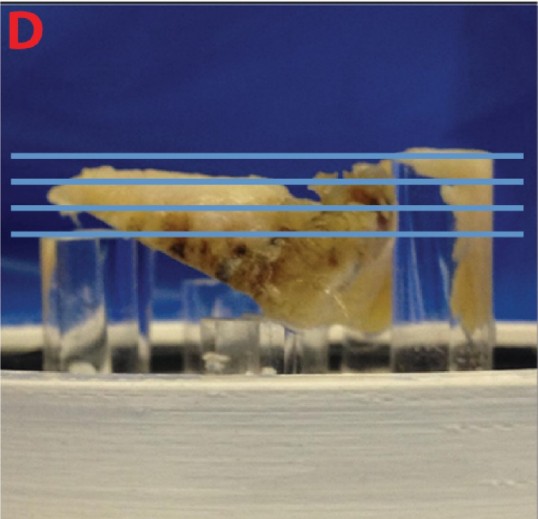

**Fig 1. Custom sample holder and SR-PCI field of view. (A)** A custom, non-metal, and waterproof sample holder was 3D-printed with an appropriate size for both samples and imaging machines and with a stable sample position with fine adjustment. **(B)** Set up for scanning PBS-submerged sample at the CLS beamline BMIT. **(C)** Top view with blue lines indicating X-ray detector field of view (55mm). **(D)** Side view with blue lines indicating how multiple X-ray beam heights (2mm) were compiled for each sample (2mm).

the Department of Anatomy, Physiology, and Pharmacology at the University of Saskatchewan. Osteoarthritic tissues were harvested from consenting patients receiving knee replacement surgery by orthopedic surgeons at Saskatoon City Hospital, Canada. All samples were stored in 10% formalin. For imaging protocols, samples were placed in a custom-designed, 3D printed sample holder (Fig 1) and submerged in phosphate-buffered saline (PBS). Imaging was performed on the medial tibial condyle of disarticulated human knee joints from four non-OA and four OA donors (Table 1). To avoid sex-dependent features that might provide

**Table 1. Tibial plateau sample characteristics.**

| Group (sample size) | Average age (distribution) | Sex | Clinical designation |
|---|---|---|---|
| **Non-OA) n = 4)** | **66.0** (59, 59, 64, 82) **yrs.** | Female | Age-appropriate, mild degeneration |
| **OA** (n = 4) | **69.5** (59, 71, 72, 76) **yrs.** | Female | Severe degeneration requiring knee replacement surgery |

Abbreviations: OA = Osteoarthritis; yrs. = years.

unnecessary variance in our statistical analyses, only female tissues were used. Also, as much as possible, cadaveric donor tissues were selected that were age-matched with osteoarthritic tissues in order to control for normal age-related changes. Average age of non-OA samples was 66.0 years, while OA samples were 69.5 years old on average (Table 1). Due to the non-destructive nature of these imaging protocols, the same sample was used for each imaging modality, providing a direct comparison of imaging modalities.

### 3T MR imaging

MRI protocols were carried out on knees scanned in the sagittal plane using an 8-channel knee-coil in the whole-body scanner 3T MRI system (Siemens 3T MRI Scanners, Germany) at the Royal University Hospital Medical Imaging Department, University of Saskatchewan. Based on a pilot study of one sample comparing T1-weighted 3D fast low-angle shot (FLASH) to dual echo steady state (DESS) MRI protocols, FLASH sequences appeared to give more consistent contrast between cartilage and surrounding tissues and so were used for this study (S1 Fig). Specific MRI parameters for FLASH sequences included: Pixel size (312.5×312.5 μm); Field of view (160×160 mm); Slice thickness (1.25 mm); Repetition time (20 msec); Echo time (7.57 msec); Number of averages (1); Echo number (1); Pixel bandwidth (130 Hz); Flip angle (12 degrees).

### SR- PCI imaging

In-line SR-PCI imaging was performed at the Biomedical Imaging and Therapy (BMIT) facility 05B1-1 beamline at the Canadian Light Source (CLS), Saskatoon, Canada. The imaging setup consisted of a double crystal bent Lau monochromator tuned to 45 keV imaging energy and a superconducting wiggler X-ray source. Samples were positioned for sagittal imaging in-line with the beam and detector on a rotating scanning stage for CT scanning. Similar to our previous data [35], tomographic data sets were collected at a 5 m sample-to-detector-distance using a beam monitor AA-60 (Hamamatsu) coupled with a Hamamatsu camera C9300-124 with a pixel size of 9x9 μm, binning data 3x3 (so effective pixel size of 27 μm x 27 μm), and an exposure time between 0.03 and 0.06 s, depending on electron ring current. The X-ray detector field of view was 55 mm, and the X-ray beam height was 2 mm (Fig 1C and 1D). Multiple scans were required to image the entire vertical height of each sample, due to the limited beam height, and these scans were stitched together using ImageJ [37].

For each data set, 2500 projections were collected over a 360˚ rotation (in order to get the entire tibial plateau in the field of view), and a set of 10 flat-field and 10 dark-field images were acquired before and after each scan. Before image reconstruction, projection corrections using flat- and dark-field images were done with an Image J macro plugin. Modified Feld Kamp Algorithm in NRecon V 1.6.10.1 (Bunker, Kontich, Belgium) was used for the non-retrieved image reconstruction to obtain image slices leading to isotropic voxels (i.e., 27 μm slice thickness) based on standard protocol [35, 40]. The two main factors of absorption and refraction contribute to in-line phase-contrast X-ray imaging. The outcomes analyzed were based upon

standard reconstruction, where the images were dominated by absorption contrast and the phase contrast contributed to edge enhancement. The images were cropped and exported into FEI Amira 6.0.1 (Oregon) for further analyses.

## Image analysis

Articular cartilage was segmented by semi-automated interactive segmentation using the CT Analyzer software package (SkyScan, Kontich, Belgium; see S2 Fig for examples of segmentation). Model-independent 3D cartilage thickness and volume mapping using a Euclidean distance transformation was also done using CT Analyzer, ultimately producing average cartilage thickness and overall cartilage volume measures, as well as a heat map of cartilage thickness. For OA samples, the surface area of the exposed subchondral bone was calculated, and each voxel in this region was assigned an average cartilage thickness value of 0 μm.

## Statistical analysis

All statistical analyzes were performed using SPSS software (v20.0. Armonk, NY: IBM Corp.). Since measurements were normally distributed, comparisons between non-OA and OA groups were performed using unpaired, two-tailed t-tests. When comparing MRI and SR-PCI data, Bland-Altman plots were used to evaluate relative differences between the imaging modalities. Ultimately, a one-sample, two-tailed t test was used to see if differences between measures were significantly different from 0, and a regression analysis tested whether the differences in measures scaled with the means. P≤0.05 was considered as statistically significant.

## Results

### MRI depicted articular cartilage less effectively than SR-PCI

To confirm the status of articular cartilage in the medical school cadaveric (non-OA) and surgical patient (OA) samples of medial tibial condyles, gross anatomical analyses were performed by an orthopaedic surgeon. These analyses revealed that samples of the non-OA group had no obvious cartilage defects, while each sample in the OA group appeared to have severe OA with exposed subchondral bone and often cartilage fibrillation (Fig 2A and 2B; also see Table 1 for sample demographics).

To evaluate any differences in resolution and contrast between the two imaging modalities, each sample was imaged by 3T MRI and SR-PCI, and virtual sections from the 3D reconstructed datasets were generated. Equivalent virtual sections from the same sample were estimated based upon known orientation of the datasets and visual cues from morphology of both the articular cartilage and adjacent subchondral bone (Fig 2C and 2D). Qualitatively, both MRI and SR-PCI could distinguish articular cartilage, subchondral bone, and surrounding fluid (i.e., PBS, which simulated synovial fluid). However, as expected from the equipment and imaging parameters used (see Methods), SR-PCI provided much higher resolution images than MRI, despite the fact that the MRI virtual section was orthogonal to the large anisotropic dimension of the MRI voxels (i.e., 1.25 mm slice thickness; Fig 2C and 2D). Also, the virtual sections from MRI showed poorer contrast at the tissue interfaces between articular cartilage and subchondral bone and between articular cartilage and surrounding fluid, compared to virtual sections from SR-PCI. Trabecular features in subchondral bone, another diagnostic of OA [30], were clearly detailed in SR-PCI, but not MRI (Fig 2C and 2D).

Segmentation of articular cartilage from the 3D image datasets allowed relative levels of cartilage thickness to be indicated with color heat maps, showing clear differences between experimental groups and also between imaging modalities. For example, OA samples had multiple

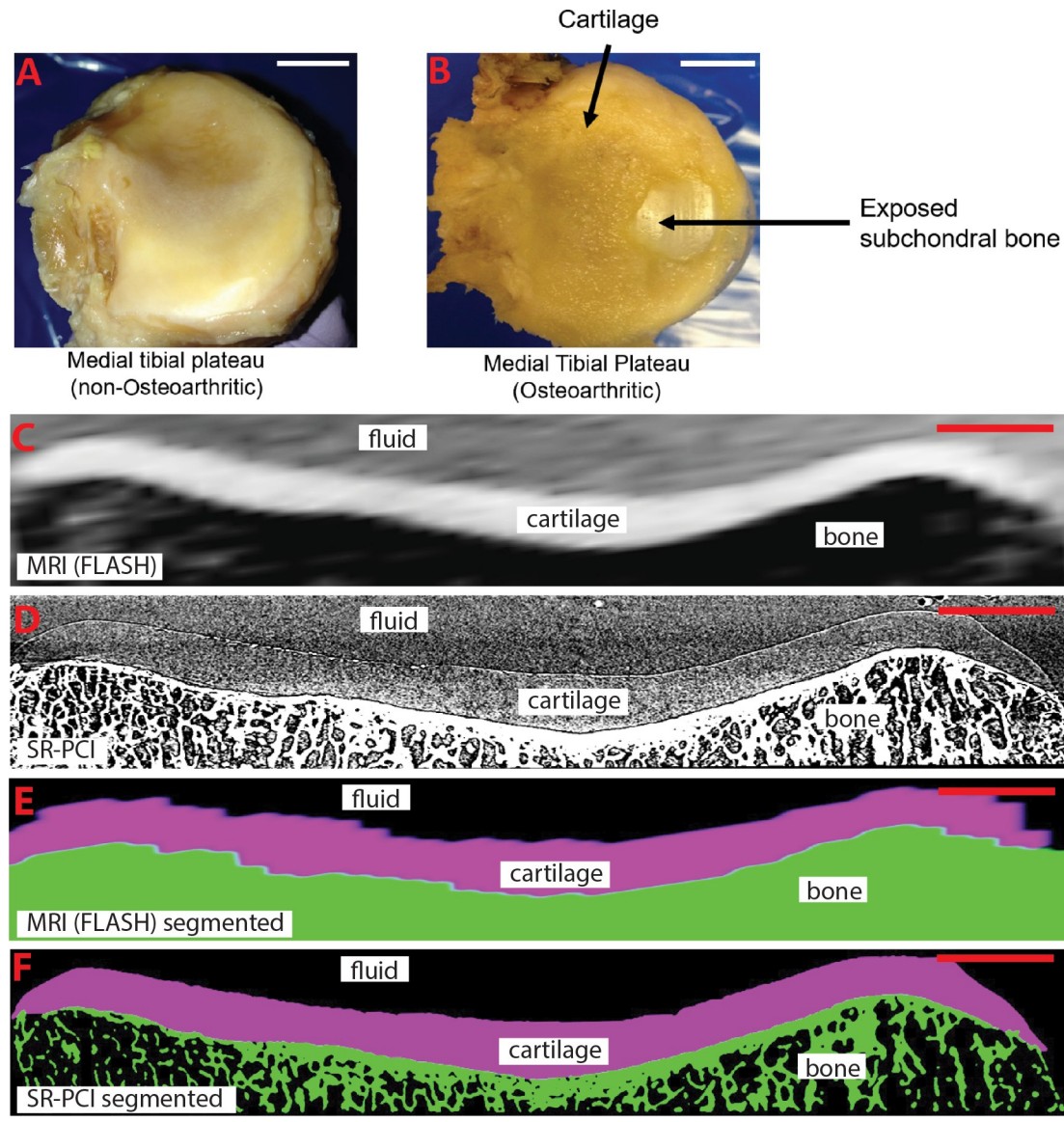

**Fig 2. Gross images of representative samples, and virtual slices and segmentations from 3D reconstructions of the same sample from MRI and SR-PCI. (A, B)** Gross images of medial tibial plateaus demonstrate normal articular cartilage appearance of a non-OA sample (A), while cartilage fibrillation and exposed subchondral bone are observed in an OA sample (B). **(C, D)** Virtual slices of equivalent regions of the same sample, based upon known orientation of the datasets and visual cues from morphology of both the articular cartilage and adjacent subchondral bone, from 3D reconstructed MRI (C) and SR-PCI (D) datasets suggest that SR-PCI has higher resolution than MRI. **(E, F)** Cartilage and bone segmentations on the same virtual slices as in panels C and D from 3D reconstructed MRI (E) and SR-PCI (F) datasets. Scale bars: A,B = 1cm; C-F = 0.5cm.

areas of severe articular cartilage thinning (red) or absence, particularly where the medial tibial condyle articulated with the medial femoral condyle, compared to non-OA samples (Fig 3). Also, compared to MRI, SR-PCI color maps of articular cartilage thickness were much less pixelated with gradual, as opposed to abrupt, changes in coloration, again consistent with the much higher resolution of SR-PCI (27 μm x 27 μm x 27 μm), compared with MRI (312 μm x 312 μm x 1250 μm (see Methods); Fig 3). In these surface views, the anisotropic dimension of the MRI voxel was evident. In total, these results led to the conclusion that SR-PCI generated more effective depictions of articular cartilage than clinical MRI.

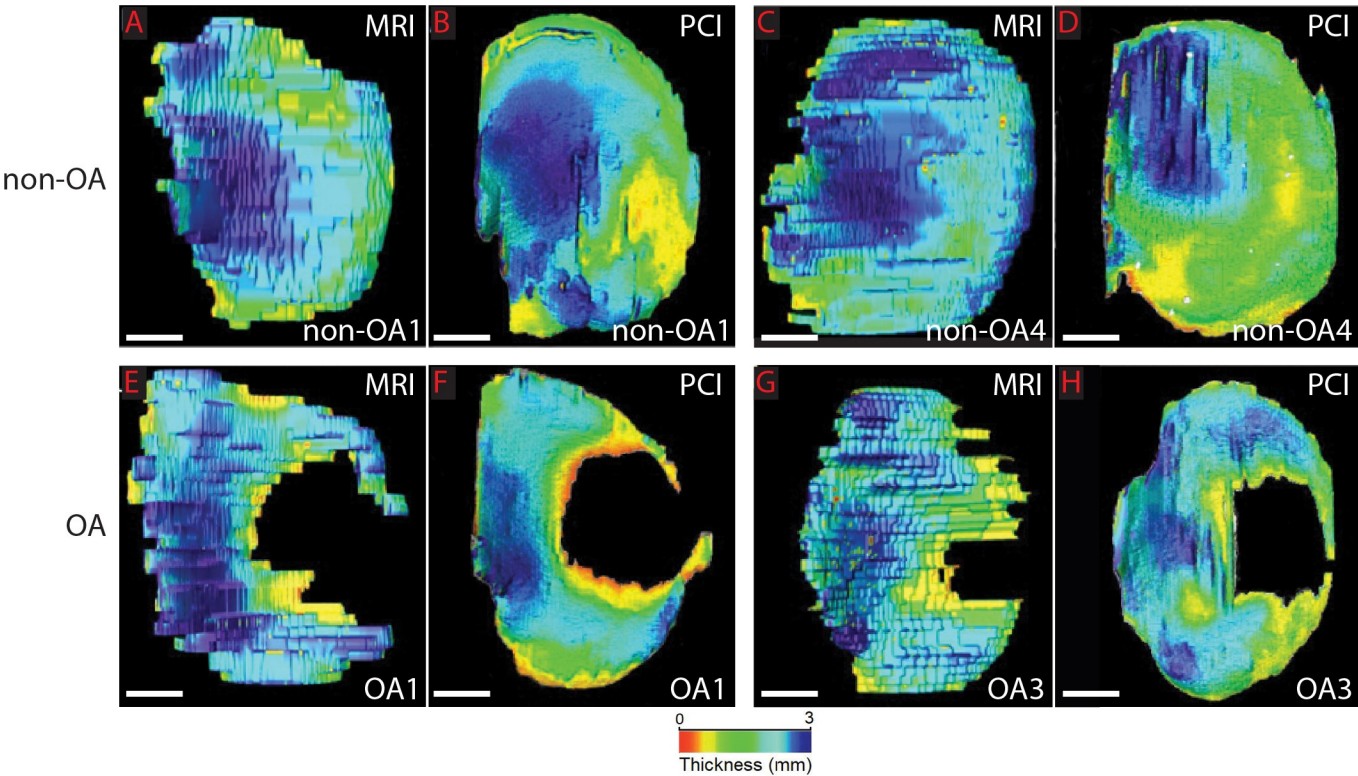

**Fig 3. Representative heat-mapped images showing cartilage thickness. (A-D)** After image segmentation of articular cartilage from medial tibial condyles, regional cartilage thickness distribution (blue = thickest, red = thinnest) in representative non-OA samples suggested that MRI estimated higher values, compared to PCI. **(E-H)** A similar trend was observed in representative OA samples, which also contained large areas without any detectable cartilage. Scale bars: A-H = 1cm.

## Both MRI and SR-PCI revealed significant decreases in articular cartilage parameters of OA samples, compared to non-OA samples

For quantitative comparisons, average cartilage thickness and overall cartilage volume were calculated from segmented regions of the 3D datasets from both 3T MRI and SR-PCI. Histological Safranin O staining of one sample confirmed the identity of cartilage in the segmented region and also provided validation of the thickness measures (S2 Fig; Table 2). Both imaging modalities revealed that the average articular cartilage thickness was significantly decreased in the OA group, compared to the non-OA group (Fig 4A and 4B). For MRI data, the non-OA average thickness was 2280 ±157 µm, while the OA average thickness was 1831 ±179 µm (p = 0.009; Table 2). For SR-PCI data, the non-OA average cartilage thickness was 2078 ±210 µm, whereas the OA measured 1648 ±243 µm (p = 0.037). Accordingly, average cartilage thickness decreased in OA samples by 19.7% for MRI data and 20.7% for SR-PCI data.

Similarly, both imaging modalities demonstrated significantly reduced overall articular cartilage volume in the OA group, compared to the non-OA group (Fig 4C and 4D). For MRI data, the non-OA cartilage volume was 2.446 ±0.195 ml, compared to 1.698 ±0.143 ml for the OA group (p = 0.001; Table 2). For SR-PCI data, the non-OA cartilage volume was 2.126 ±0.469 ml, while the OA group measured 1.417 ±0.262 ml (p = 0.039). The decrease in overall cartilage volume of OA samples was 30.6% for MRI data and 33.4% for SR-PCI data. In summary, analyses of articular cartilage from MRI and SR-PCI data indicated that both imaging methods revealed significantly reduced average cartilage thickness and overall cartilage volume in the OA group, compared to the non-OA group.

**Table 2. Sample measurements.**

| Group | Samples | MRI: Average cartilage thickness* (µm) | SR-PCI: Average cartilage thickness* (µm) | MRI: Overall cartilage volume (ml) | SR-PCI: Overall cartilage volume (ml) |
|---|---|---|---|---|---|
| Non-OA | 1 | 2290 | 1879 | 2.476 | 2.139 |
| | 2 | 2054 | 1931 | 2.674 | 2.758 |
| | 3 | 2389 | 2173 | 2.200 | 1.963 |
| | 4 | 2386 | 2328 | 2.432 | 1.643 |
| Non-OA average (± SD) | | 2280 (±157) | 2078 (±210) | 2.446 (±0.195) | 2.126 (±0.469) |
| OA | 1 | 1646 | 1454 | 1.785 | 1.027 |
| | 2 | 1725 | 1588 | 1.792 | 1.578 |
| | 3 | 1915 | 1546 | 1.728 | 1.563 |
| | 4 | 2039 | 2002 | 1.488 | 1.498 |
| OA average (± SD) | | 1831 (±179) | 1648 (±243) | 1.698 (±0.143) | 1.417 (±0.262) |

*includes 0 values for voxels over the exposed subchondral bone surface area in OA samples

## MRI over-estimated articular cartilage parameters, compared to SR-PCI

While both MRI and SR-PCI were able to detect significant differences in articular cartilage parameters between severe OA and non-OA, they did not show equivalent measures for each parameter in each group. Compared to SR-PCI, average cartilage thickness and overall

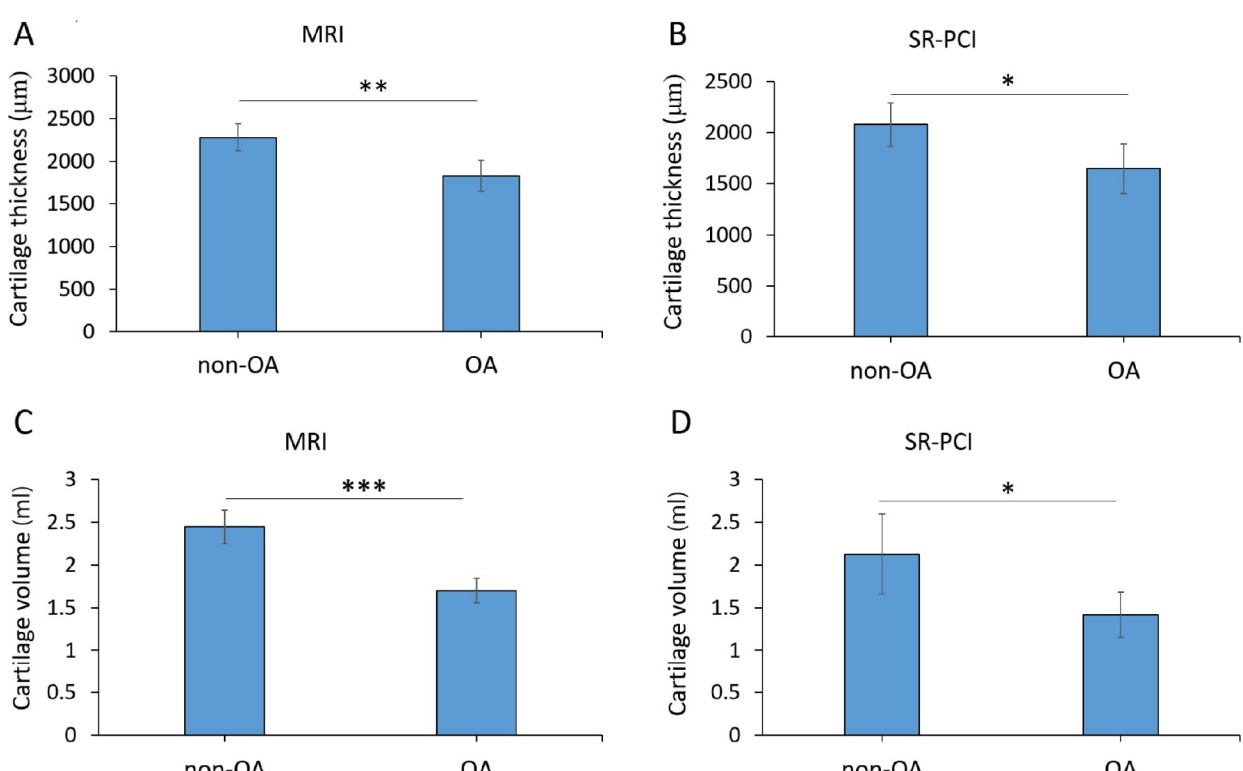

**Fig 4. Both MRI and SR-PCI demonstrated significant decreases in clinically-relevant articular cartilage parameters. (A, B)** Quantitative analyses of average cartilage thickness (µm) showed that OA samples were significantly thinner than non-OA samples from both MRI (A, **p = 0.009) and SR-PCI data (B, *p = 0.037). **(C, D)** Quantitative analyses of overall cartilage volume (ml) showed that OA samples had significantly less cartilage than non-OA samples from both MRI (C, ***p = 0.001) and SR-PCI data (D, *p = 0.039).

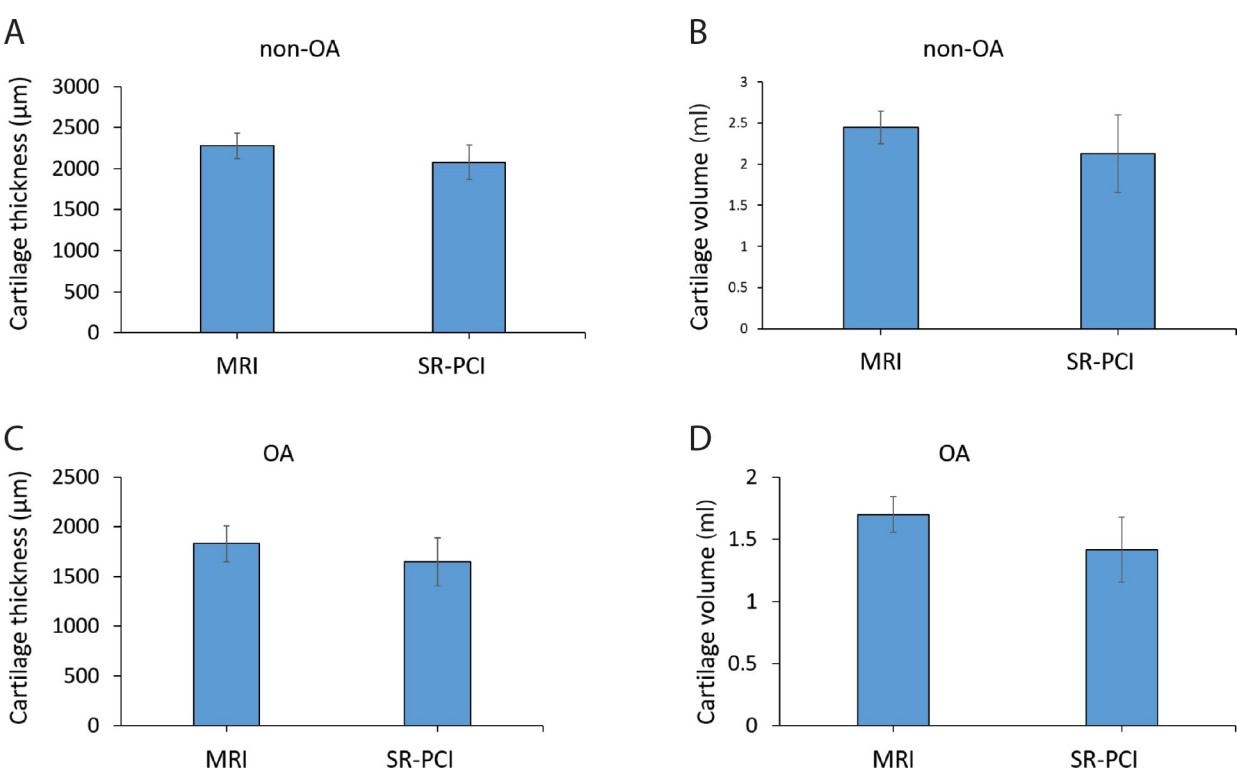

**Fig 5. Comparisons of imaging modalities with only one experimental group were inconclusive. (A, B)** Quantitative analyses of average cartilage thickness (A) and overall cartilage volume (B) of the non-OA samples trended towards an increased measure in MRI, compared to SR-PCI, but the results were not statistically significant. **(C, D)** Quantitative analyses of average cartilage thickness (C) and overall cartilage volume (D) of the OA samples trended towards an increased measure in MRI, compared to SR-PCI, but the results were not statistically significant.

cartilage volume trended higher in MRI, although the differences were not statistically significant when considering each group separately (Fig 5). Measurements from MRI data produced an average thickness of articular cartilage in non-OA samples that trended 9.7% higher ($p = 0.078$) than those from SR-PCI data (2280 ±157 μm and 2078 ±210 μm, respectively; Table 2). For OA samples, measurements from MRI data produced an overall cartilage volume that trended 11.1% higher ($p = 0.078$) than those from SR-PCI data (1831 ±179 μm and 1648 ±243 μm, respectively; Table 2). Regarding overall cartilage volume in non-OA samples, measurements from MRI data trended 15.1% higher ($p = 0.174$) than those from SR-PCI data (2.446 ±0.195 ml and 2.126 ±0.469 ml, respectively; Table 2). For OA samples, measurements from MRI data trended 19.8% higher ($p = 0.188$) than those from SR-PCI data (1.698 ±0.143 ml and 1.417 ±0.262 ml, respectively; Table 2).

Bland-Altman analyses across both non-OA and OA groups were used to compare dissimilarities between MRI and SR-PCI in obtaining two clinically-relevant measurements of articular cartilage. Simply combining non-OA and OA sample data did not find significant differences between MRI and SR-PCI measurements, likely because of the wide variation in measurements between non-OA and OA samples within each imaging modality. Bland-Altman plots reflected that MRI datasets calculated larger values for both average cartilage thickness and overall cartilage volume than SR-PCI datasets (Fig 6A and 6C). Indeed, the differences in MRI and SR-PCI measurements of each sample's average cartilage thickness ($p = 0.005$) and overall cartilage volume ($p = 0.033$) were statistically greater than zero. MRI data for average cartilage thickness was 11.0% (±8.2%) higher than SR-PCI data, while overall

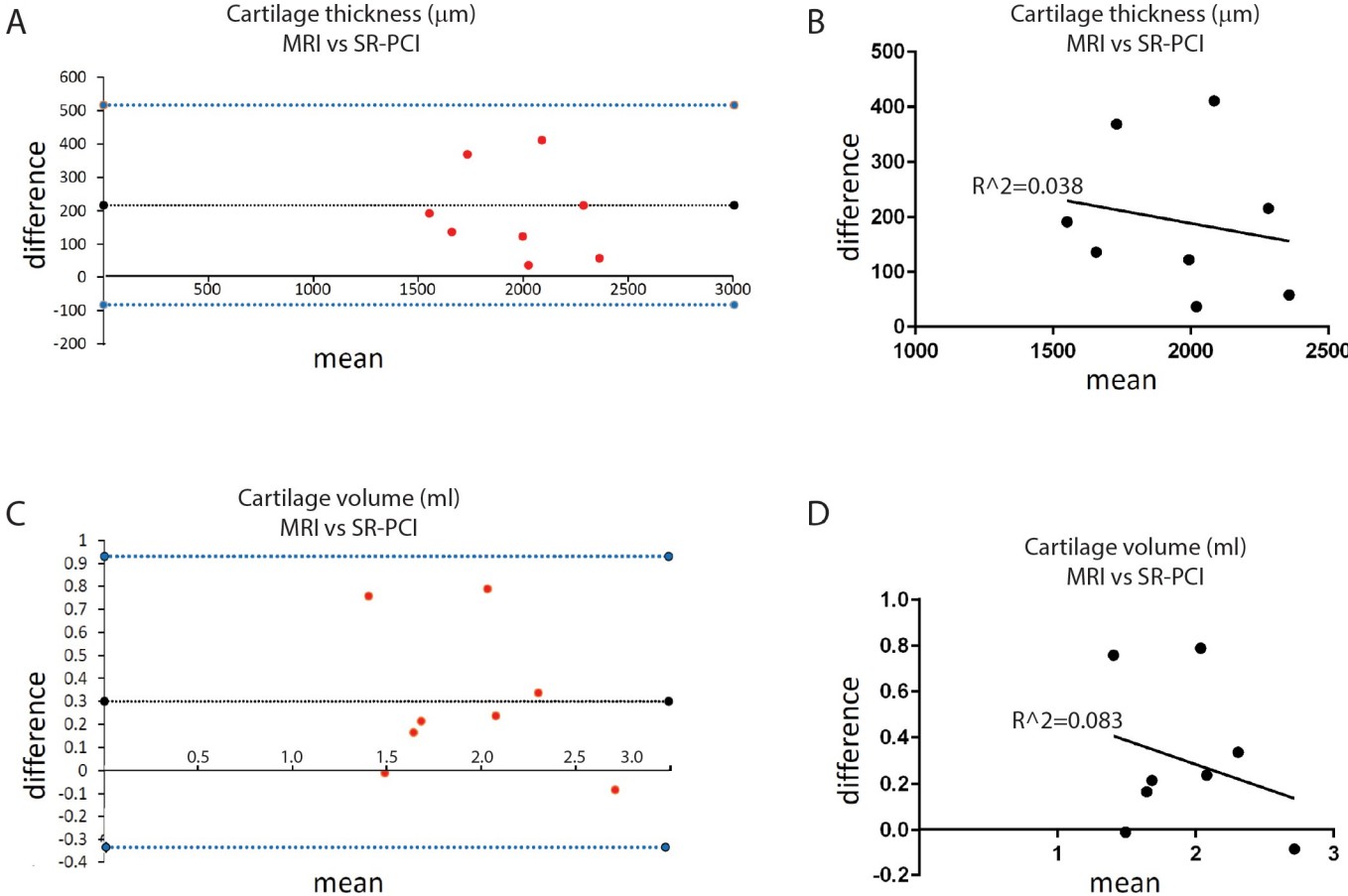

**Fig 6. MRI significantly over-estimated clinically-relevant articular cartilage parameters, compared to SR-PCI. (A)** Bland-Altman analyses of average cartilage thickness suggested that MRI over-estimated measurements, compared to SR-PCI; this finding was verified as significant by a one-sample t-test of differences compared to 0 (p = 0.005). **(B)** Regression analysis confirmed that the difference between the measures from MRI and SR-PCI was independent of the scale of the measurement (p = 0.646; R^2 = 0.038). **(C)** Bland-Altman analyses of overall cartilage volume suggested that MRI over-estimated measurements, compared to SR-PCI; this finding was verified as significant by a one-sample t-test of differences compared to 0 (p = 0.033). **(D)** Regression analysis confirmed that the difference between the measures from MRI and SR-PCI was independent of the scale of the measurement (p = 0.489; R^2 = 0.083). For Bland-Altman plots (A,C), the x-axis is labelled at difference = 0, dotted black line represents the mean difference among samples, and dotted blue lines represent 95% confidence intervals from the mean difference.

cartilage volume was 21.2% (±26.3%) higher for MRI data than SR-PCI data. Finally, regression analyses confirmed that the difference between measures of cartilage thickness (p = 0.646; R^2 = 0.038) or cartilage volume (p = 0.489; R^2 = 0.083) was not significantly related to the scale of the measures (Fig 6B and 6D). In total, these data clearly demonstrated that, compared to SR-PCI, MRI over-estimates clinically-relevant parameters of articular cartilage health when imaging samples *ex vivo*.

## Discussion

Because insufficient biomarkers exist for the early diagnosis of OA, new imaging approaches to improve the diagnosis of OA are needed. Such improvements also will improve the ability to assess progression of this complex disease, predict final joint damage, and monitor the effectiveness of experimental therapies [1, 7].

Clinical MRI has been adapted for quantitative morphological assessment of articular cartilage, notably to reveal average cartilage thickness and overall cartilage volume. However, the

resolution of clinical MRI might not offer the capability to delineate details of articular cartilage, meaning that early cases of OA might go undiagnosed [28, 29, 41]. Recent imaging innovations, such as grating-based X-ray PCI-CT (not in-line) and SR-PCI, have been highlighted in proof-of-principle studies for high-resolution evaluation of clinically-relevant articular cartilage parameters [24, 30, 42, 43].

Here, we provided for the first time a direct comparison of MRI and SR-PCI techniques on multiple non-OA and OA human articular cartilage samples, using image segmentation to quantitate average cartilage thickness and overall cartilage volume. By assessing morphological features of cartilage that may be related to function, these imaging techniques make it possible to distinguish between healthy and OA cartilage. While 3D-DESS MRI protocols are often a sequence of choice for quantitative measurements of articular cartilage (e.g., [44]), our pilot imaging suggested more consistent contrast between cartilage and surrounding tissues using the FLASH protocol (i.e., cartilage had lighter pixels than both the fluid and bone). A future study using DESS images would address this potentially influential decision. Both MRI and SR-PCI revealed that the OA group had significantly lower average cartilage thickness and overall cartilage volume, compared to the non-OA group. The OA samples in this study were very severe cases, since they were recovered from knees of patients undergoing total knee replacement surgeries. In fact, articular cartilage in all OA samples had been depleted so much as to expose large areas of subchondral bone. Perhaps if the OA samples had less severe loss of articular cartilage, then both imaging modalities might not have revealed significant decreases. Future studies with less severe cases of OA will better discriminate whether 3T MRI or SR-PCI can diagnose early OA more accurately.

Qualitative evaluations of articular cartilage images demonstrated an increased ability of SR-PCI to distinguish tissue boundaries, compared to MRI. Virtual sections from the 3D image reconstructions illustrated that SR-PCI defined better the boundaries between bone and articular cartilage or between articular cartilage and surrounding fluid. Such enhanced edge contrast is a characteristic feature of PCI, since the image reconstructions incorporate phase diffraction shifts from one medium to another [38]. Additionally, edge contrast in MRI images could suffer from a chemical-shifts artifact, due to different water and fat compositions of various joint tissues [41]. In MRI, gradient imaging and fat suppression algorithms can be used to saturate the excitation of fat protons, which may improve contrast resolution between cartilage and free fluid structures, such as synovial fluid, or even help identify subchondral bone cysts [41]. Changes to subchondral bone are also characteristic of OA [1, 2], so the ability of SR-PCI to provide fine detail of subchondral bone might also be advantageous for the early diagnosis of OA.

Quantitative evaluations of articular cartilage images in the study groups showed a significantly higher average thickness and overall volume from MRI data, compared to SR-PCI data, which is likely due to the difference in imaging resolution strengths of the two modalities. Color map visualization of cartilage thickness suggested that SR-PCI could provide a better description of morphological features, including cartilage erosion, than clinical MRI. Individual articular cartilage parameters also trended higher for each experimental group in measurements from MRI data, compared to those from SR-PCI. Bland-Altman analyses confirmed a significant increase in both average cartilage thickness and cartilage volume in measurements from MRI data compared with SR-PCI. Overestimation of cartilage thickness and volume are probably related to reduced imaging spatial resolution in MRI, compared to SR-PCI, which then could cause errors in subsequent measurements. Unexpectedly, despite decreased resolution of MRI, the standard deviations of MRI measures were lower. For example, standard deviation was 10% of the average cartilage thickness for MRI images of OA samples, while it was 15% for SR-PCI images. Given that MRI overestimated measures, compared to SR-PCI, we

speculate that partial volume effects contributed to the decreased standard deviation of MRI, especially in thinner regions of articular cartilage.

Surface views of the cartilage showed the anisotropic and relatively large (1.25 mm) slice thickness component of the MRI voxel (running top-to-bottom in Fig 3 images), which was a major factor in the dramatically decreased resolution of MRI. However, since this dimension was orthogonal to the cross-section views (see square pixels in Fig 2C), it likely affected measures of overall cartilage volume much more than average cartilage thickness measures. Overall, the nominal resolution of these 3T MR images would be 312.5 μm x 312.5 μm x 1250 μm voxel, while the nominal resolution of SR-PCI would be 27 μm x 27 μm x 27 μm voxel (see Methods). Therefore, compared to MRI, SR-PCI had several orders of magnitude better resolution (20 x $10^3$ μm$^3$ vs. MRI's 12.2 x $10^7$ μm$^3$), even that being limited in principle by the detector resolution and settings employed. As the superficial layer of articular cartilage is approximately 300 μm thick, loss of this layer and the beginnings of OA could be present, but not detectable with MRI [28, 45–47].

So, was 3T MRI or SR-PCI more accurate in revealing clinically-relevant articular cartilage parameters? Unfortunately, no gold standard exists for answering this question with these samples, so direct comparison between the actual measurements and those from MRI or SR-PCI is not easy. The articular cartilage measurements reported here for both imaging modalities, such as average cartilage thickness of 2280 μm (from MRI) or 2078 μm (from SR-PCI), in the medial tibial condyle of non-OA specimens were comparable to previous studies [7, 19, 24, 30, 39]. Histological preparations might be a good gold standard, but they also have a measurement error due to processing, which may cause shrinkage due to specimen dehydration or other structural changes that occur during embedding, sectioning, and staining of the tissues. While imperfect as a gold standard, our histological sections provided further support for the articular cartilage measures reported here for these imaging modalities. Future test-retest scans and evaluating inter-reader variability for the semi-automated segmentation would increase evaluation of accuracies of MRI and SR-PCI in depicting articular cartilage. The potential question of which modality is preferred, however, must be based on the clinical indication and question. MRI is good at evaluating the gross morphology and structure of the cartilage. SR-PCI could be better at visualization of not only the interface between cartilage and bone, but also fine cartilage details, such as surface morphology and minor defects that could lead to disease progression.

At least for the severe cases of OA examined here, these data suggest that overall cartilage volume might be a better diagnostic of articular cartilage health than average cartilage thickness. Regardless of the imaging modality employed, OA measures for average cartilage thickness were 20% lower than non-OA measures. Greater decreases in overall cartilage volume were measured by both imaging modalities, with OA measures over 30% lower than non-OA measures. A future study would be useful evaluating the influence of different available cartilage thickness measurement algorithms on similar analyses as performed here.

SR-PCI might provide dramatically improved cartilage imaging, compared to 3T MRI, but unfortunately, implementing any synchrotron-based imaging for clinical applications is currently impractical. Synchrotrons are not commonly available, and they are extremely complicated and expensive to establish, requiring expertise across science and engineering disciplines and massive budgets. In addition, protocols would need to be developed to address safety concerns about radiation dose to patients and potential movement of patients during scanning. Therefore, SR-PCI still has limited clinical applications currently, but preclinical animal studies are prime targets for synchrotron-based imaging. Large animal studies, such as pig, can replicate gross features of human articular cartilage disease progression and are useful in evaluating preclinical therapeutic strategies [48]. Also, a recent development in the field is to image

living animals using synchrotron-based techniques, even imaging the same animal multiple times, potentiating longitudinal studies of OA progression from early stages [37, 49–53].

## Conclusion

Taken together, these findings clearly demonstrated that SR-PCI provided high-resolution images that may improve assessment of clinically relevant articular cartilage parameters, such as cartilage thickness and overall volume. In addition, SR-PCI can highlight fine details of the cartilage surface, interfaces between cartilage and bone, and even fine details of the subchondral bone structure. Comparing 3T MRI to SR-PCI, both MRI and SR-PCI revealed that the OA group had significantly lower average cartilage thickness and overall cartilage volume, compared to the non-OA group. However, MRI data showed a significantly higher average thickness and overall volume, compared to SR-PCI data, which is likely due to the difference in imaging resolution strengths of the two modalities. As such, with further development, SR-PCI might increase accurate evaluation of morphological changes in articular cartilage not only for early diagnosis of OA disease, but also for monitoring disease progression and response to treatment.

## Supporting information

**S1 Fig. Pilot comparison of MRI sequences suggested that FLASH gave more consistent contrast at cartilage boundaries than DESS. (A, B)** Virtual slices of similar regions of the same sample from 3D reconstructed MRI datasets using either FLASH (A) or DESS (B) imaging sequences suggested that contrast between the cartilage and overlying fluid and subchondral bone was more consistent with FLASH. Scale bars: A,B = 0.5cm.
(EPS)

**S2 Fig. A histological stain of a sectioned sample shows a similar cartilage thickness as calculated from 3D reconstructed image analyses.** A tissue section of a non-OA sample was stained with Safranin O, independently confirming the presence of cartilage, meanwhile providing independent support for the measures of cartilage thickness obtained from the segmented cartilage on virtual slices. Scale bar = 0.4cm.
(EPS)

## Acknowledgments

We thank the selfless donors of cadavers to the Body Bequeathal Program in the Department of Anatomy, Physiology, and Pharmacology, and also appreciate the donors of surgically removed OA tissues during total knee arthroplasty at City Hospital of Saskatoon. Thanks to Bianca Sarkis for helping prepare samples for histology and to Shreyas Jois, Aditya Manek, and the University of Saskatchewan College of Medicine Histology Core Facility.

## Author Contributions

**Conceptualization:** L. Dean Chapman, David M. L. Cooper, B. Frank Eames.

**Data curation:** Suranjan Bairagi, Oghenevwogaga J. Atake, B. Frank Eames.

**Formal analysis:** Suranjan Bairagi, Mohammad-Amin Abdollahifar, David M. L. Cooper, B. Frank Eames.

**Funding acquisition:** B. Frank Eames.

**Investigation:** Suranjan Bairagi, Oghenevwogaga J. Atake, B. Frank Eames.

**Methodology:** Suranjan Bairagi, M. Adam Webb, Ning Zhu, David M. L. Cooper, B. Frank Eames.

**Project administration:** B. Frank Eames.

**Resources:** William Dust, Sheldon Wiebe, George Belev, L. Dean Chapman, M. Adam Webb, Ning Zhu, David M. L. Cooper, B. Frank Eames.

**Supervision:** B. Frank Eames.

**Validation:** Mohammad-Amin Abdollahifar.

**Visualization:** Oghenevwogaga J. Atake.

**Writing – original draft:** Mohammad-Amin Abdollahifar, B. Frank Eames.

**Writing – review & editing:** William Dust, Sheldon Wiebe, M. Adam Webb, Ning Zhu, David M. L. Cooper, B. Frank Eames.

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
