## [Decision Letter · Decision Letter 0]

13 Feb 2023

PONE-D-22-34378Clinical MRI overestimates articular cartilage thickness and volume compared to synchrotron radiation phase-contrast imagingPLOS ONE

Dear Dr. Eames,

Thank you for submitting your manuscript to PLOS ONE. After careful consideration, we feel that it has merit but does not fully meet PLOS ONE’s publication criteria as it currently stands. Therefore, we invite you to submit a revised version of the manuscript that addresses the points raised during the review process.

 Reviewer #2 has raised important concerns about the study design and the MRI protocol used in this study. Please address these comments with special cares. 

We look forward to receiving your revised manuscript.

Kind regards,

Zhentian Wang, Ph.D.

Academic Editor

PLOS ONE

Journal Requirements:

"This study was supported by grants to BFE: Royal University Hospital (RUH) Foundation, the Saskatchewan Health Research Foundation (SHRF) Establishment Grant, and the Canadian Institutes of Health Research (CIHR) project grant 148683.  We thank the selfless donors of cadavers to the Body Bequeathal Program in the Department of Anatomy, Physiology, and Pharmacology, and also appreciate the donors of surgically removed OA tissues during total knee arthroplasty at City Hospital of Saskatoon.  Part of the research described in this paper was performed at the Canadian Light Source, a national research facility of the University of Saskatchewan, which is supported by the Canada Foundation for Innovation (CFI), the Natural Sciences and Engineering Research Council (NSERC), the National Research Council (NRC), CIHR, the Government of Saskatchewan, and the University of Saskatchewan."

7. We note that you have indicated that data from this study are available upon request. PLOS only allows data to be available upon request if there are legal or ethical restrictions on sharing data publicly. For more information on unacceptable data access restrictions, please see http://journals.plos.org/plosone/s/data-availability#loc-unacceptable-data-access-restrictions.

Reviewers' comments:

Reviewer's Responses to Questions

**Comments to the Author**

1. Is the manuscript technically sound, and do the data support the conclusions?

Reviewer #1: Yes

Reviewer #2: Partly

2. Has the statistical analysis been performed appropriately and rigorously? 

Reviewer #1: Yes

Reviewer #2: I Don't Know

3. Have the authors made all data underlying the findings in their manuscript fully available?

Reviewer #1: Yes

Reviewer #2: No

4. Is the manuscript presented in an intelligible fashion and written in standard English?

Reviewer #1: Yes

Reviewer #2: Yes

5. Review Comments to the Author

Reviewer #1: Accurate evaluation of the morphological characteristics of articular cartilage is essential for the early detection and diagnosis of osteoarthritis (OA). Currently, 3T magnetic resonance imaging (MRI) is widely used to detect OA in clinic. Meanwhile, synchrotron radiation phase-contrast imaging computed tomography (SR-PCI), as a unique imaging technique, is capable of high-resolution imaging of biological soft tissues, including articular cartilage. In this study, the authors evaluated human articular cartilage quantitatively based on MRI and SR-PCI simultaneously. Compared with the non-OA group, the average cartilage thickness and cartilage volume in the OA group were significantly reduced by both imaging methods. In addition, the high resolution SR-PCI is able to map the cartilage surface more accurately than MRI, thus providing more accurate information on cartilage degeneration in OA.

In general, the paper is well written and flows well. I have but a few comments:

1. Keywords- The format of all keywords should be further unified.

2. Introduction- In addition to MRI and SR-PCI, are there any other common techniques used to conduct imaging studies of articular cartilage? Perhaps the authors can supplement the research status of this part in the introduction.

3. Figure 2- Can the authors add corresponding scalebars to each panel in Figure 2?

4. Figure 3- The background of Figure 3H should be consistent with other subgraph images.

5. Figure 5- Even if there is no statistical significance, it is suggested that the author should also give the corresponding P value to facilitate readers' better understanding.

6. The author mentions in the discussion that histological preparations might be a good gold standard, but they also have a measurement error due to processing, which may cause shrinkage due to specimen dehydration or other structural changes that occur during embedding, sectioning, and staining of the tissues. Since histology is the gold standard, should the authors consider including histological results for comparison with SR-PCI and MRI results?

Reviewer #2: The aim of this manuscript is to compare two imaging methods for cartilage thickness measurement, MRI and synchrotron radiation phase-contrast imaging. The topic is up-to-date as since many years the OA community has been searching for reliable (imaging) biomarkers that would be helpful in disease diagnosis and treatment monitoring. Cartilage thickness has been identified as a potential candidate for a reliable OA biomarker; therefore, it is important to investigate on its properties, specificity and sensitivity. Present study tries to address some of the pitfalls of cartilage thickness measurement from MRI images.

Unfortunately, the results of this study does not provide any considerable answers to the research questions, mostly because of an unfortunate study design. Manuscript contains substantial methodological flaws, which need to be addressed before publication.

1. The MRI part is referred to as “clinical” throughout the manuscript. As the ex-vivo cartilage samples have been examined, the expression “clinical” should be omitted. Using a clinical scanner with the clinical protocol is not sufficient to label this study as clinical.

2. The choice of SR-PCI as a counter-method for MRI is unclear. Authors even admitted that it could not serve as a gold standard measurement of cartilage thickness. Moreover, it is not mentioned in the manuscript how the slices were matched. What is the slice thickness of SR-PCI images? SR-PCI, despite superior resolution compared to MRI, is also just an approximation of the cartilage thickness measurement. Even on histological slices, it is hard to define the cartilage thickness mostly due to ragged bone-cartilage interface. Any other imaging modality with the lower resolution suffers from inaccuracies in thickness measurement simply because the resolution does not allow seeing those details.

3. That being said, the title “The Clinical MRI overestimates articular cartilage thickness and volume compared to synchrotron radiation phase-contrast imaging” is misleading, as it implies that SR-PCI is a gold standard method.

4. Introduction, last paragraph: here, the research question should be clearly stated. Was it the comparison between two modalities. Or was it a feasibility study for SR-PCI? Please omit the sentences discussing the results: “While both imaging modalities revealed significant differences..”, “These results suggest a specific means to improve..”.

5. Conclusion: Currently it is a compilation of the general statements. Please state the main finding(s) of your study here.

6. Another serious issue is the MRI protocol, which is far away from state-of-the art of the protocol for volumetric cartilage analysis (including the thickness measurement). The “OA Research Society International recommendation” to use T1-3D-GRE applies to focal cartilage defect detection, rather than volumetric analysis. On the contrary, the contrast of this sequence is very bad for volumetry analysis due to low contrast between cartilage and a surrounding tissue (bone, fluid effusion, fat) – it is obvious also on the Figure 2 (C). Isotropic 3D-DESS is currently a sequence of choice for any volumetric measurements. Using T1-3D-GRE penalizes the MRI part of this study to a large extent.

7. It is also recommended to investigate the influence of different available cartilage thickness measurement algorithms (nearest neighbor, surface normal, local thickness, field lines, etc), as they may provide different results for different imaging modalities.

8. In the results section, it is surprising that the MRI-thickness SD is lower compared to SR-PCI (6.8% vs 10% in non-OA and 9.8% vs 14.8%). Is that a mean +/- SD over 4 samples per group? Or is that a mean value through the slices? Either way, considering the superior image resolution of SR-PCI, one would expect much lower SD for SR-PCI measurements. In principle, the cartilage thickness measured here from MRI images includes 3 to 4 pixels from cartilage surface to the bone, at given resolution the accuracy is expected to be rather low.

9. Please add a representative cartilage segmentation in high detail for both modalities

10. To be able to discriminate between the accuracies of thickness measurement of these two methods (without having a ground truth), it is recommended to perform test-retest scans (including removing the specimen from the scanner) and inter-reader variability for the semi-automated segmentation.

11. Maybe a subject for a discussion: looking at the Fig 4 and Fig 5, these results suggest that using volume a primary endpoint for the OA detection is much better than using cartilage thickness.

6. PLOS authors have the option to publish the peer review history of their article (what does this mean?). If published, this will include your full peer review and any attached files.

Reviewer #1: No

Reviewer #2: No

---

## [Author Response · Author response to Decision Letter 0]

19 Jul 2023

please see Response to Reviewers .docx attached to the revised documents submitted

---

## [Decision Letter · Decision Letter 1]

10 Aug 2023

PONE-D-22-34378R1MRI overestimates articular cartilage thickness and volume compared to synchrotron radiation phase-contrast imagingPLOS ONE

Dear Dr. Eames,

Thank you for submitting your manuscript to PLOS ONE. After careful consideration, we feel that it has merit but does not fully meet PLOS ONE’s publication criteria as it currently stands. Therefore, we invite you to submit a revised version of the manuscript that addresses the points raised during the review process.

We look forward to receiving your revised manuscript.

Kind regards,

Zhentian Wang, Ph.D.

Academic Editor

PLOS ONE

Journal Requirements:

Reviewers' comments:

Reviewer's Responses to Questions

**Comments to the Author**

1. If the authors have adequately addressed your comments raised in a previous round of review and you feel that this manuscript is now acceptable for publication, you may indicate that here to bypass the “Comments to the Author” section, enter your conflict of interest statement in the “Confidential to Editor” section, and submit your "Accept" recommendation.

Reviewer #1: All comments have been addressed

Reviewer #2: (No Response)

2. Is the manuscript technically sound, and do the data support the conclusions?

Reviewer #1: Yes

Reviewer #2: Yes

3. Has the statistical analysis been performed appropriately and rigorously? 

Reviewer #1: Yes

Reviewer #2: Yes

4. Have the authors made all data underlying the findings in their manuscript fully available?

Reviewer #1: Yes

Reviewer #2: Yes

5. Is the manuscript presented in an intelligible fashion and written in standard English?

Reviewer #1: Yes

Reviewer #2: Yes

6. Review Comments to the Author

Reviewer #1: I think authors have revised the manuscript with my comments. The current manuscript has more related information and the mistakes have been solved. I therefore agreed this manuscript can be accepted for publication.

Reviewer #2: The authors' effort put in improving the manuscript is appreciated. Most of the comments have been addressed.

However, there are still couple of remaining issues concerning favoring FLASH over DESS sequence.

1. The first issue is the DESS contrast in Supplemental Figure 1. That does not resemble the contrast of properly optimized DESS. See below the example of the clinical DESS implementation.

Here is the example of a clinical DESS contrast: https://imgtr.ee/image/DESS-clinical.Wb7Fo you see it is substantially different to what is presented in SFig. 1. This should be addressed.

The second issue is the different resolution of DESS and FLASH - why is that? Both sequences should be set up in the same way.

Moreover, after adding the SFig.1, it seems that the FLASH image in the SFig.1 is visually different to the FLASH image in Fig. 2. Please clarify which contrast has been used. If that in the Fig. 2, it certainly disqualifies MRI.

These issues can falsely lead the readers to the thinking that DESS performs poorly in terms of segmentation as it has worse contrast and resolution than FLASH - and that is not correct.

Additionally, what is the speculation "This unexpected difference might be due to the dissected samples used, instead of larger samples or whole patient limbs." based on? Why would be the cartilage contrast be different between in-vivo and ex-vivo conditions?

2. Referring to the comment #7, I still believe that the different algorithm should be mentioned at least in the discussion as a limitation, as they might contribute to the thickness uncertainty quite a lot.

7. PLOS authors have the option to publish the peer review history of their article (what does this mean?). If published, this will include your full peer review and any attached files.

Reviewer #1: No

Reviewer #2: No

---

## [Author Response · Author response to Decision Letter 1]

23 Aug 2023

please see Response to Reviewers file

---

## [Editor Report · Decision Letter 2]

5 Sep 2023

MRI overestimates articular cartilage thickness and volume compared to synchrotron radiation phase-contrast imaging

PONE-D-22-34378R2

Dear Dr. Eames,

We’re pleased to inform you that your manuscript has been judged scientifically suitable for publication and will be formally accepted for publication once it meets all outstanding technical requirements.

Kind regards,

Zhentian Wang, Ph.D.

Academic Editor

PLOS ONE
---

## [Editor Report · Acceptance letter]

25 Sep 2023

PONE-D-22-34378R2 

MRI overestimates articular cartilage thickness and volume compared to synchrotron radiation phase-contrast imaging 

Dear Dr. Eames:

I'm pleased to inform you that your manuscript has been deemed suitable for publication in PLOS ONE. Congratulations! Your manuscript is now with our production department. 

Kind regards, 

on behalf of

Prof. Zhentian Wang 

Academic Editor

PLOS ONE